# The Different Faces of *Arabidopsis arenosa*—A Plant Species for a Special Purpose

**DOI:** 10.3390/plants10071342

**Published:** 2021-06-30

**Authors:** Żaneta Gieroń, Krzysztof Sitko, Eugeniusz Małkowski

**Affiliations:** Plant Ecophysiology Team, Faculty of Natural Sciences, University of Silesia in Katowice, 28 Jagiellońska Str., 40-032 Katowice, Poland; zgieron@us.edu.pl

**Keywords:** *Arabidopsis arenosa*, hyperaccumulation, autopolyploidy

## Abstract

The following review article collects information on the plant species *Arabidopsis arenosa*. Thus far, *A. arenosa* has been known as a model species for autotetraploidy studies because, apart from diploid individuals, there are also tetraploid populations, which is a unique feature of this *Arabidopsis* species. In addition, *A arenosa* has often been reported in heavy metal-contaminated sites, where it occurs together with a closely related species *A. halleri*, a model plant hyperaccumulator of Cd and Zn. Recent studies have shown that several populations of *A. arenosa* also exhibit Cd and Zn hyperaccumulation. However, it is assumed that the mechanism of hyperaccumulation differs between these two *Arabidopsis* species. Nevertheless, this phenomenon is still not fully understood, and thorough research is needed. In this paper, we summarize the current state of knowledge regarding research on *A. arenosa*.

## 1. *Arabidopsis arenosa*—General Information

*Arabidopsis arenosa*, previously known as *Cardaminopsis arenosa*, is a species of flowering plants in the family *Brassicaceae*, which includes two subspecies: *A. arenosa* ssp. *arenosa* and *A. arenosa* ssp. *borbasii* [1,2]. *A. arenosa* is a model plant species used to study the natural phenomenon of autopolyploidyzation, which means the duplication of the whole genome within one species, in contrast to allopolyploids, which are the result of genome duplication via interspecies hybridisation. *A. arenosa* is more closely related to its diploid sister taxa *A. lyrata* and to *A. halleri* (metal hyperaccumulator), than to *A. thaliana*. Although *A. arenosa* and *A. halleri* are closely related, studies which compare both species are still scarce [2,3,4,5]. A distinguishing feature of this species is its natural occurrence both in a diploid (2n = 2C = 16) and tetraploid (2n = 4C = 32) form, in contrast to *A. halleri*, in which only diploid forms (2n = 2C = 16) were found [6,7,8,9].

*A. arenosa* grows in subfertile soils. It has been observed in sandy areas, dry grasslands and roadsides. Plants of this species are present in many sites in Europe; however, their location differs depending on the level of ploidy, because tetraploid forms were noted in Slovenia, France and Belgium to the west and north of the Carpathians, while diploid forms have been found in South-East Europe, the Balkan Peninsula and northern Hungary. Moreover, the Carpathian mountain arch, in particular the western Carpathians, is one of the two contact zones where both diploid and tetraploid populations co-exist. The second contact zone of both cytotypes is in the Slovenian Forealps [7,10,11,12]. *A. arenosa*, similarly to *A. halleri*, is a pseudo-metallophyte [13,14,15,16], which is the species used to study adaptations to the environments that are highly contaminated with heavy metals [3,4,17,18]. Both species can be commonly found on metalliferous and non-metalliferous sites [18,19,20,21,22]. The pictures of both diploid and tetraploid forms of *A. arenosa* are presented in Figure 1.

*A. arenosa* is a biennial or short-lived perennial herb. The leaves are arranged in a rosette at the ground level, and no runners form from the rosette. The shape of the leaf blade is obovate to oblanceolate, pinnatifid to lyrate-pinnatifid and with three to eleven lateral lobes on each side. The stems of this plant species are erect, simple, or with few or more branches from the base. *A. arenosa* produces tiny flowers whose petals are white to lavender, obovate, obtuse or cut at the apex [2]. The siliques of *A. arenosa* are always smooth and raised up, in contrast to *A. halleri*, in which a narrowing reflecting the seeds occurs (Figure 2). Moreover, the pedicles in *A. arenosa* are at an acute angle to peduncle, whereas in *A. halleri* they are almost perpendicular (Figure 2). The period of flowering and fruit production is from April to July or, rarely, to August.

## 2. Adaptation to Autopolyploidy

*Arabidopsis arenosa* exists in nature as diploid and autotetraploid populations. The ability to duplicate the whole genome of autotetraploid populations of *A. arenosa* is closely linked to the adaptation of tetraploids to duplication without errors in meiosis; thus, most importantly, avoiding the formation of multivalents or univalents, which cause sterility [23,24,25]. Tetraploid individuals of *A. arenosa* perform diploid-like chromosome pairings at meiosis, where bivalents are selected randomly from four homologs [8,12,26]. A comparative analysis of the genomes of diploid and autotetraploid populations allowed one to distinguish a group of 44 genes with divergent selection between ploidy levels, responsible for the ability of *A. arenosa* to perform stable meiosis and, consequently, to create subsequent generations of autotetraploid populations of this species. This set of genes mainly affects the meiotic crossover initiation pathway, but is also involved in other functions, such as chromosomal cohesion or segregation, chromatin structure, DNA repair and transcriptional regulation [8,12,23,26]. This set includes *ASYNAPTIC 1* and *3* (*ASY1* and *ASY3*) genes, which encode core components of the chromosome axis and present a strong signature of adaptive evolution focused on a mutation that changes a single amino acid [8,12,24]. Other genes in this important group are *SISTER CHROMATID COHESION2* (*SCC2*), responsible for encoding the adherin loading cohesin during meiosis [26,27], as well as *STRUCTURAL MAINTENANCE OF CHROMOSOMES 3*, *5* and *6* (*SMC3*, *SMC5* and *SMC6*), genes related to meiosis, which show signatures of selective sweeps [12,26,28]. Moreover, the meiotic genes show the strongest features of introgression resistance in tetraploid populations; that is, increased divergence between cytotypes, with commitment limited diversity in tetraploids, and tetraploid monophy in the contact zones of both cytotypes. It follows from the necessity of exclusion of the introduction of diploid-like meiotic alleles to tetraploids, which would result in the formation of multivalents [29,30].

The ability to duplicate the whole genome, and thus the ability of autotetraploidyzation, has certain consequences. Results of a recent study support the possibility that it is possible to overcome the adaptation obstacles caused by reduced selection efficiency, due to increased mutational input [31,32]. This is confirmed by the results of comparative genome analyses of diploid and tetraploid populations, which show that in tetraploid populations, even the increased masking of beneficial mutations is not sufficient to slow the adaptation process, due to the higher number of non-synonymous polymorphisms fixed by positive selection [29]. Additionally, it has been shown that autopolyploidy may increase genome flexibility, allowing plants to adapt to more heterogeneous conditions. As a result, tetraploid populations of *A. arenosa* inhabit sites polluted by human activities, going beyond the natural areas inhabited by diploid ancestors [25,29,33].

## 3. *Arabidopsis arenosa* and Heavy Metals

*Arabidopsis arenosa* is considered to be a pseudo-metallophyte, that is, a plant species that inhabits both metalliferous and non-metalliferous sites [3,18,22]. For many years, this species has been considered as an excluder, a plant that is able to survive on metalliferous soils, maintaining a physiological content of zinc (Zn) and low cadmium (Cd) in its above-ground tissues in plants growing in situ [6,34]. Nevertheless, the first reports began to appear about the ability of several populations of *A. arenosa* to hyperaccumulate Zn [4,22,35], according to the generally accepted definition proposed by Van der Ent and colleagues [36]. The term hyperaccumulator is defined as plants that are able to take up and accumulate a specific concentration of heavy metals from the soil. It should also be emphasised that in hyperaccumulators, heavy metals should be easily transported from the root to the shoot for their accumulation in above-ground organs without any observable symptoms of phytotoxicity being displayed [14,34,36,37,38,39]. The hyperaccumulation of Zn was first described for *Nocceaea caerulescens*, also from the family Brassicaceae, in 1865 [40]. Currently, approximately 721 plant species have been reported that show HM hyperaccumulation. This number accounts for 0.2% of all known plant species, and the new hyperaccumulator species are being reported [34,36,37,40,41,42,43]. From this group of plants, Cd and Zn hyperaccumulation was shown mainly in the Brassicaceae family, and in a few other species from other families, for example, Crassulaceae (*Sedum alfredii*, *Sedum plumbizincicola*) [44,45,46]. In previous studies, Nadgórska-Socha et al. [47] showed that the content of metals such as Cd, Zn, Pb, Fe and Mn is considerably higher in *A. arenosa* compared to *Plantago lanceolata* and *Plantago major* grown in the metal-contaminated soils in situ. Moreover, in *A. arenosa*, the translocation factor (the ratio between the metal concentration in shoots to the metal concentration in roots) was above 1 for Cd and Zn, which suggested the hyperaccumulation of both metals [47]. Recent studies confirmed the ability of *A. arenosa* to hyperaccumulate Cd and Zn [35]. In this study, they showed a higher ability to hyperaccumulate Zn than Cd in *A. arenosa*, as this feature for Zn has been proven in five out of six studied metallicolous populations. In contrast, Cd hyperaccumulation was found only in three metallicolous populations [35]. In addition, the adaptation of another two populations of *A. arenosa* from southern Poland to grow in HM-contaminated soil was shown. However, the capability for the hyperaccumulation of Zn or Cd by both populations was not presented [3].

Populations of *A. arenosa* in various habitats show specific morphological features. For example, the length of the leaves of plants grown on a copper (Cu) mining heap was 2.5 times smaller than the leaves of plants grown in non-contaminated soil. By contrast, the length of the roots of the seedlings from the heap was remarkably longer compared with the plants from the reference site [15,48]. Similarly, an *A. arenosa* population growing on a Zn/Pb waste heap had smaller size, thicker and narrower leaves, with fewer trichomes. Moreover, the root test showed a higher tolerance to Cd, Zn and Pb for the population from the heap compared to the reference population [18]. On the other hand, both the metallicolous (M) and non-metallicolous (NM) *A. arenosa* tetraploid plants growing in hydroponic solution without HMs displayed higher above-ground biomass compared to the *A. halleri* plants [22]. However, in Cd-containing media, a more significant decrease in biomass was observed in the NM population than in the M population of *A. arenosa* [22]. Root biomass also decreased by treatment with Cd by less than 50% in the M population and about 90% in the NM population compared to the control [22]. Similarly, the Zn treatment caused root growth limitation in both *A. arenosa* and *A. halleri* [4]. The presence of Zn can also increase the volume and root length of the hyperaccumulator plant, which has been shown for *S. alfredii* [49].

In general, photosynthesis in hyperaccumulator plants has been seldom investigated. As a result, the resistance of photosynthetic apparatus to the toxic effect of metals in this group of plants is poorly understood [50,51,52], in comparison with large amount of data on crop plants [53,54,55,56,57,58]. The studies of the photosynthetic apparatus parameters of *A. arenosa* in situ demonstrated the adaptability and high level of tolerance of the metallicolous population to HM. Nevertheless, NM populations had better PSII energy fluxes parameters compared to the M populations. However, the values of the parameters studied for a population from the extremely polluted area were closer to the NM populations than to the most M populations [35]. This may result from a considerable variation in the resistance of the photosynthetic apparatus to heavy metals between populations from metalliferous sites. In the case of *A. arenosa* plants from highly polluted sites in Piekary Śląkie (Poland), the exceptionally high resistance of the photosynthetic apparatus to metals was found. The photosynthesis parameters were similar to those in plants from the reference sites. However, such high resistance is not observed in all populations from metalliferous sites [35]. In *N. caerulescens*, Cd and Zn were accumulated mostly in the vacuoles of epidermal cells; in consequence, the metals were non-toxic for PSII. Exposure to 800 μM Zn or 40 μM Cd in a hydroponic experiment increased Ca^2+^ translocation to the above-ground parts and increased Fe^3+^ uptake as a PSII photoprotective mechanism [59]. A similar mechanism was found in *A. halleri*, but HMs were accumulated in the vacuoles of leaves’ parenchyma [60]. A different reaction was demonstrated by Morina and Küpper [52] for *A. halleri* treated with Cd, who found that Cd is mainly accumulated in the veins and reduces the distribution of Fe and Zn from the veins. However, no effect on the distribution of Ca was found. Thus, the disturbance in the leaf nutrient homeostasis after Cd treatment could be the main factor behind the progressive inhibition of the PSII reaction centers and the decrease in quantum yield of the electron transport [52]. There are no data about the mechanism of sequestration of HMs in *A. arenosa* leaves. It has been shown that the metallicolous population of *A. arenosa* exhibit similar values of chlorophyll *a* fluorescence to the metallicolous *A. halleri* population: the HM hyperaccumulator. A significantly higher chlorophyll content index in the metallicolous and non-metallicolous *A. arenosa* populations compared to the *A. halleri* populations was also found. Moreover, the higher content of this pigment in *A. arenosa* compared to the metallicolous *A. halleri* population may indicate a better physiological status of this *A. arenosa* population [22].

Polyphenols, such as flavonols and anthocyanins, are generally recognized as molecules involved in stress protection in plants and have multiple functions in acclimation processes to an excessive amount of HMs [61,62]. It has been demonstrated in *A. thaliana* that anthocyanins play a major role in protecting against metal stresses [63]. Muszyńska et al. [64] showed that the enhanced accumulation of phenolic acids provides an efficient neutralization of metal-induced ROS in metallicolous ecotypes of *Alyssum montanum*. Furthermore, the accumulation of flavonols in leaves was a characteristic reaction of M-ecotypes of *A. montanum* during HM treatment [64]. Moreover, the higher level of anthocyanins content index has been reported in *A. arenosa* populations from metalliferous rather than non-metalliferous sites (Figure 3), indicating the increased tolerance of metallicolous populations to the toxic effects of HM [22,35]. However, the values of this parameter for M and NM *A. arenosa* populations were lower than for the M population of *A. halleri* [22]. Moreover, flavonols and anthocyanins have been shown to contribute significantly to the response to HM in hyperaccumulating and non-hyperaccumulating plant species, leading to enhanced metal antioxidant and chelating capacity [65]. The general comparison of metallicolous and non-metallicolous populations of *A. arenosa* is presented in Figure 3.

Despite the growing scientific interest in *A. arenosa* species, the mechanism of coping with HMs in the natural habitats of this species is still unknown. Even though *A. arenosa* often occurs in the same sites with *A. halleri*, comparative studies of both species have shown contrasting metal accumulation strategies. In *A. arenosa,* only metallicolous populations exhibited Zn and Cd hyperaccumulation, while this trait was observed in both metallicolous and non-metallicolous populations of *A. halleri* [17,22,35,66]. The molecular studies indicated that HM hyperaccumulation is associated with a change in the expression level of numerous genes (Table 1). The first stage is the uptake and transport of a metal by the roots. The primary role in this process is played by members of the ZIP family (the zinc-regulated transporter/iron-regulated transporter-like proteins), whose expression is high in the roots and/or shoots of hyperaccumulating plants [45,46,67]. ZIP19 and ZIP23 transporters are mainly responsible for the uptake of zinc by the roots of *A. halleri* and *N. caerulescens* [46]. In non-hyperaccumulator species, the expression of several ZIP genes is low and increases during Zn deficiency [46,67]. In addition, it was found that Cd treatment induces the higher expression of genes related to Cd uptake and transport in roots (*IRT1*, *ZIF1*) and shoot (*ZIF1* and *YSL3*), as well as Cd vacuolar sequestration (*HMA3*) [68]. IRT1 (iron-regulated transporter 1) encoding a low selective Fe-uptake transporter in the root epidermis is also involved in Zn, Cd and/or nickel (Ni) uptake [67]. Therefore, it has been shown that different expression levels of these genes in *A. halleri* are associated with the differential accumulation of these metals in shoots [68,69]. Subsequently, Zn enters the cortex with the participation of ZIP4 and/or IRT1, and next to the endodermis through IRT3 and ZIP5, ZIP19, ZIP23, transporters [45,46,70]. Due to the presence of Casparian strips, the further transport of Zn is only possible via ZIP4/ZNT1 transporters. The increased expression of genes encoding these transporters was demonstrated in the roots of both *A. halleri* and *N. caerulescens* [46,67,71]. The next stage is the loading of metals into the xylem, which is mediated by the HM transporter, the HMA4 protein, a plasma membrane ATPase [40,72]. In *A. halleri*, *HMA4* occurs in three copies and shows higher expression (four- to ten-fold) compared to *HMA4* in *A. thaliana*. In *A. halleri*, this higher expression of *HMA4* is crucial for the process of hyperaccumulation and hypertolerance [46,67,71,73]. All three copies of HMA4 allow xylem metal loading and distribution to leaves due to the activity of *A. halleri* in vascular tissues. This pump also provides a metal exclusion from sensitive tissues such as the root tip [67,73]. The increased expression of *HMA4* has been demonstrated not only for *A. halleri* but also for other hyperaccumulators, including *N. caerulescens*, *S. alfredii* and *S. plumbizincola* [74,75,76]. The xylem loading process is also carried out by other transporters, such as YSL (yellow stripe-like protein), which are mainly responsible for the transport of Zn, Cu, manganese (Mn), Ni, Cd and Fe. In addition, the YSL proteins take part in the long-distance transport of metals in the xylem together with the protein FRD3 (ferric reductase defective 3). However, the FRD3 transporter is only responsible for the Fe/Zn translocation in the xylem [46,77,78,79]. After the metals reach the above-ground organs of a plant, they enter the leaf cells through ZIP4 and ZIP6 proteins [46]. Both proteins were located in the plasma membrane of the *A. halleri* shoots and *S. alfredi* shoots and roots [46,80,81]. In leaves, metals are sequestered and detoxified in vacuoles, where their toxic effects are limited. They reach the vacuole by MTP1 proteins (metal tolerance protein 1). In the hyperaccumulator *A. halleri*, the presence of five copies of *MTP1* gene was found, which is strongly expressed both in shoots and roots [67,71,82,83]. The high expression of this gene has also been detected in other hyperaccumulators such as *N. caerulescens* and *S. alfredii* [84,85]. Other important transporters responsible for metals entering into the vacuoles is HMA3. In *A halleri*, a higher expression level of *HMA3* was detected in the mesophyll, whereas *N. caerulescens* had higher expression in the bundle sheath of the veins [46,79,86]. Transporters from the HMA family also show a detoxifying function in other hyperaccumulators such as *S. alfredii* and *S. plumbizincola* [46,79,87,88,89]. Another family of genes involved in vacuole sequestration is the NRAMP gene family (natural resistance-associated macrophage protein). The high expression of *NRAMP1, NRAMP3, NRAMP4,* and *NRAMP5* has been detected in *A. halleri* and *N. caerulescens* [46,47,79,81,90,91,92,93,94]. In *A. halleri*, the higher expression of the transporter CAX1 (cation-exchanger 1) also seems to be responsible for Cd hypertolerance [95,96].

Table 1 shows that the function of a large number of genes responsible for metal uptake, xylem leading, long-distance transport or vacuole sequestration have been identified in *A. halleri* and *N. caerulescens*. By contrast, in *A. arenosa,* the expression level of genes related to metal uptake or hypertolerance are still unknown (Table 1). These knowledge gap should be addressed, particularly that several hyperaccumulating Zn and Cd populations of *A. arenosa* have recently been found [35]. The first paper, which compares the tolerance to heavy metals in *A. arenosa* and *A. halleri* at a genomic level, was published by Preite et al. [3]. They investigated in both species the populations inhabiting the same metalliferous and non-metalliferous sites. Despite the relatively close relationship between *A. arenosa* and *A. halleri*, a modest degree of gene and functional network convergence between species was demonstrated. The comparison between metallicolous and non-metallicolous populations of *A. arenosa* identified five candidate genes exhibiting selective sweep signatures convergent between both types of populations. These genes are: PHT5;1 (vacuolar Pi sequestration), AT1G71210 (pentatricopeptide repeat (PPR) superfamily protein), AXY8 (altered xyloglucan 8; 1,2-a-L-fucosidase), NRPC2 (nuclear RNA polymerase C2), and AT4G19050 (NB-ARC domain-containing disease resistance protein). Surprisingly, none of these genes are responsible for metal uptake or tolerance. Thus, it is evident that further research should be conducted on genes exhibiting selective sweep signatures as well as those connected with metal uptake and tolerance (Table 1).

## 4. Metal Tolerance and Interaction with Soil Microorganism Communities

Fungi interacting with many plant species has a beneficial effect in the adaptation of plants to various types of environmental stresses, which has been described many times [106,107,108,109,110]. *Arabidopsis arenosa* was recognized as a non-mycorrhizal species in early studies. Similarly, many other Brassicaceae family species did not show the ability of symbiosis with mycorrhizal fungi [110,111]. Nevertheless, it has been shown that both *A. arenosa* and *A. halleri* from a serpentine soil occasionally showed the penetration of arbuscular mycorrhizal hyphae into the cortex, but vesicles and arbuscules were not formed [112].

Moreover, studies were conducted on the interaction of the endophytic fungus (*Mucor* sp.) isolated from *A. arenosa* from the mine wastelands. The studies on plants treated with metals showed several important beneficial aspects of fungus presence, such as improving the water and phosphorus status. The fungus also increased the fresh weight almost two-fold compared with the control. The plants inoculated with Mucor sp. and growing on the mine dump substrate had a three times higher N content in the shoots than the uninoculated plants [110,113]. Additionally, interaction with the fungus led to the upregulation of many genes responsible for ethylene metabolism, which resulted in a significant elongation of root hairs. Moreover, the improved transport of Zn, Cd and Fe from root to shoot was noted in inoculated plants. This indicates that the interaction with the fungus has a beneficial effect on the management and distribution of toxic metals in plant tissues to minimize the harmful effects in the roots and detoxification in the shoots [110]. Furthermore, the genetic–biochemical diversity was measured in rhizosphere soil of *A. arenosa* and *A. halleri* by denaturing gradient gel electrophoresis (PCR-DGGE). It was found that biodiversity indices in metal-contaminated soil differed between both species and was lower in the *A. halleri* rhizosphere [114].

## 5. Conclusions and Prospects for the Future

*Arabidopsis arenosa* has gained interest among scientists for its unique feature of duplicating the whole genome of autotetraploid populations while also existing as a diploid form. Due to this feature, this species has become a model plant for research on the autopolyploidization process. However, this plant species is also able to grow and develop on soils highly contaminated with HM. Moreover, the hyperaccumulation of Zn and Cd has been found recently in several tetraploid populations of *A. arenosa* from Poland. Further investigations on the mechanisms of hyperaccumulation of both metals in this plant species are necessary, particularly in countries other than Poland. At present, only tetraploid populations are known as hyperaccumulators. Thus, it is necessary to find out if diploid populations can also grow on metal-contaminated soils and hyperaccumulate Zn and/or Cd. Although *A. arenosa* is well known as a pseudo-metallophyte, our knowledge on the mechanism of metal uptake and tolerance in this plant species is very poor, in contrast to *A. halleri* or *N. caerulescens*. Thus far, to the best of our knowledge, no research studies have been published for *A. arenosa,* which present the expression level of such genes or gene families as *HMA2*, *HMA3*, *HMA4*, ZIP, NRAMP and MTP, which are crucial for metal tolerance and/or hyperaccumulation. Furthermore, it was found that the tetraploid population from the non-metalliferous site had considerably lower resistance to Cd compared to the tetraploid metallicolous population. These results show that tetraploidy in *A. arenosa* is not sufficient for high tolerance to metal toxicity. Thus, investigations are necessary to find out if metal hyperaccumulation and/or tolerance are connected with higher specific gene copy numbers, as shown for *A. halleri* or *N. caerulescens*. The above knowledge can be further exploited in high biomass plant species that could be used for phytoremediation or in the production of fortified food.

## Figures and Tables

**Figure 1 plants-10-01342-f001:**
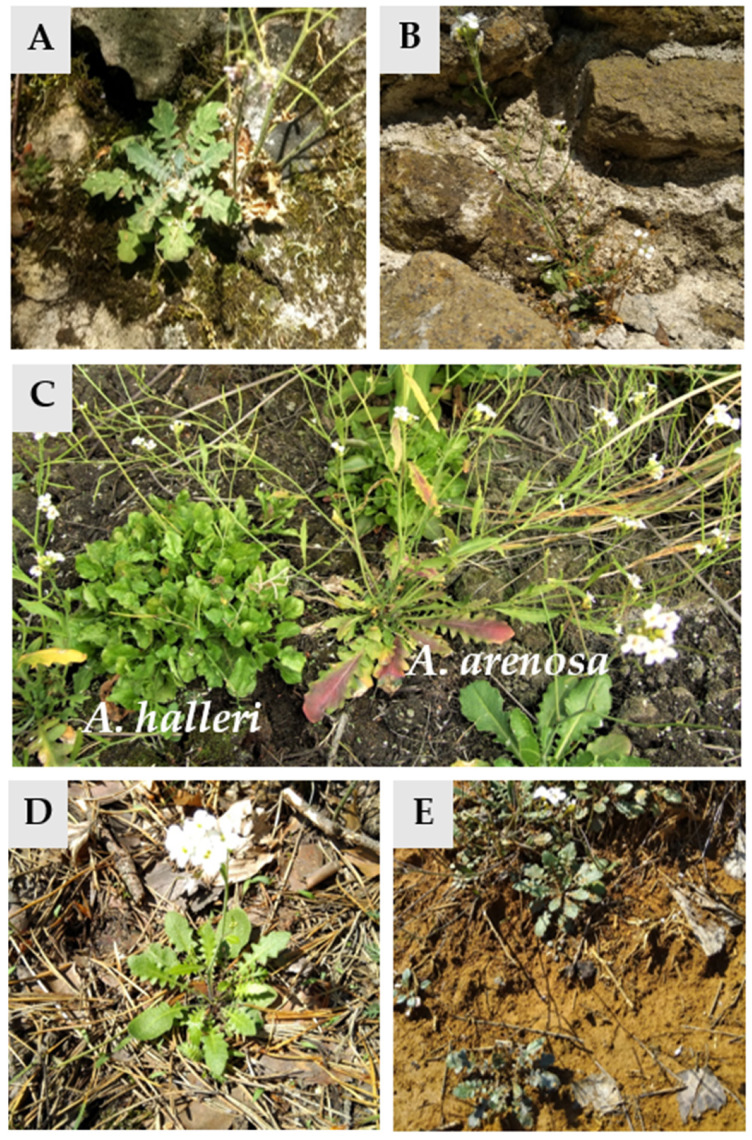
Pictures of *Arabidopsis arenosa* in situ. (**A**)—diploid (2C) from Csesznek, Hungary; (**B**)—diploid (2C) from Szigliget, Hungary; (**C**)—comparison of *A. arenosa* (4C) and *A. halleri* (2C) in Piekary Śl., Poland; (**D**)—tetraploid (4C) from Klucze, Poland; (**E**)—tetraploid (4C) from Dołki, Poland.

**Figure 2 plants-10-01342-f002:**
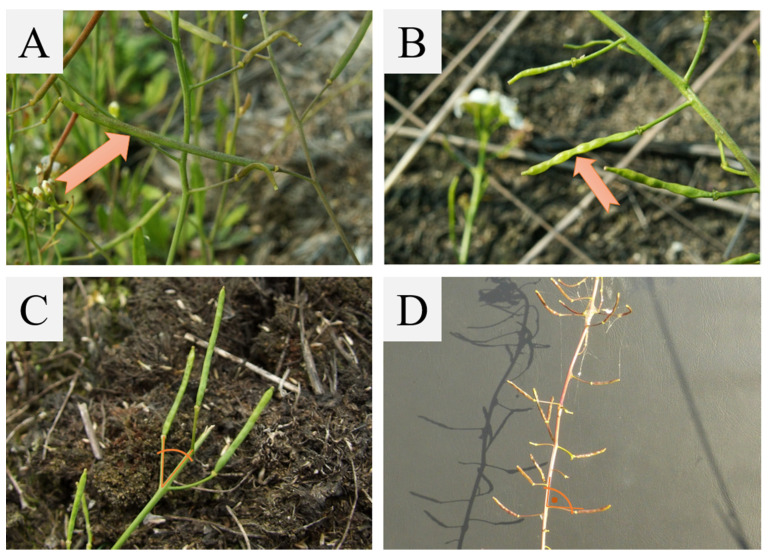
Pictures of the siliques of *A. arenosa* (**A**,**C**) compared to the siliques of *A. halleri* (**B**,**D**). On picture (**B**), the arrow indicates a characteristic narrowing that is not present in *A. arenosa* siliques. Pictures (**C**,**D**) show the differences in the location of the pedicles relative to the peduncles in both species.

**Figure 3 plants-10-01342-f003:**
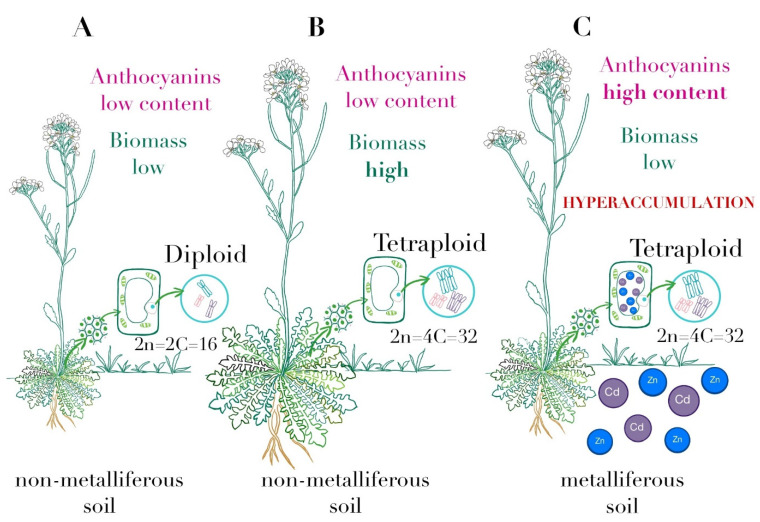
Several features that differ between the populations of *A. arenosa*. (**A**) Diploid populations from non-metalliferous sites are characterized by a low anthocyanins content and low biomass. (**B**) Tetraploid populations from non-metalliferous sites are characterized by low anthocyanins content and high biomass. (**C**) Tetraploid populations from metalliferous sites show a high content of anthocyanins and low biomass; additionally, they have the ability to hyperaccumulate heavy metals.

**Table 1 plants-10-01342-t001:** Genes putatively assigned functional roles in mechanisms of heavy metal hyperaccumulation and hypertolerance in *Arabidopsis halleri* (Ah), *Noccaea caerulescens* (Nc) and *Arabidopsis arenosa* (Aa).

Gene Name	Annotation	Function	Species	Ref
*ZIP4*	ZIP family, Zn transporter	Metal uptake in cells	Ah, Nc, Aa?	[71,80,81]
*ZIP5*	ZIP family, Zn transporter		Ah, Nc, Aa?	[71,80,81]
*ZIP6*	ZIP family, Zn transporter		Ah, Nc, Aa?	[71,80,93]
*ZIP9*	ZIP family, Zn transporter		Ah, Nc, Aa?	[71,81,93]
*ZIP19*	ZIP family, Zn transporter		Ah, Nc, Aa?	[46,67,94]
*ZIP23*	ZIP family, Zn transporter		Ah, Nc, Aa?	[46,67,94]
*IRT1*	ZIP family, Fe^2+^ transport protein		Ah, Nc, Aa?	[69,80,97]
*IRT3*	ZIP family, Zn^2+^/Fe^2+^ transport protein		Ah, Nc, Aa?	[46,71,98]
*ZNT1*	Zn transporter in *Noccaea caerulescens*	Metals influx into cells responsible for xylem loading	Nc	[46,99,100]
*ZNT2*	Zn transporter in *Noccaea caerulescens*		Nc	[46,67]
*ZTN5*	Zn transporter in *Noccaea caerulescens*		Nc	[46,67,101]
*HMA3*	plasma membrane metal ATPase pump	Metal vacuolar sequestration	Ah, Nc, Aa?	[80,86,87,93]
*HMA4*	plasma membrane metal ATPase pump	Metal loading into the xylem	Ah, Nc, Aa?	[73,75,76,86,88]
*MTP1*	Metal tolerance protein	Metal vacuolar sequestration	Ah, Nc, Aa?	[70,82,83,102]
*YSL3*	Fe-NA transporter	Xylem loading and unloading; long-distance transport	Ah, Nc, Aa?	[68,103,104]
*YSL5*	Metal-NA transporter		Nc, Aa?	[103,104]
*YSL6*	Metal-NA transporter		Ah, Nc, Aa?	[71,104]
*FRD3*	Citrate transporter	Long-distance transport	Ah, Nc, Aa?	[70,71,77,90]
*NRAMP1*	Vacuolar metal transporter	Zn sequestration in the vacuole of leaf cells	Nc, Aa?	[46,90,93]
*NRAMP3*	Vacuolar metal transporter		Ah, Nc, Aa?	[46,90,93,105]
*NRAMP4*	Vacuolar metal transporter		Nc, Aa?	[90,93,105]
*NRAM 5*	Vacuolar metal transporter		Nc, Aa?	[46,90,91]

## Data Availability

Not applicable.

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
