# Peer review of "The Different Faces of Arabidopsis arenosa—A Plant Species for a Special Purpose"

_plants, 2021, doi:10.3390/plants10071342_

Round 1
Reviewer 1 Report
The review “The Different Faces of Arabidopsis arenosa – a Plant Species for a Special Purpose” by Gieroń et al., reports exhaustive studies on A. arenosa , a metal hyperaccumulator species that up to now received little interest if compared with A. halleri and N. caerulescens. The work is interesting and add knowledge to the metal hyperaccumulator species. However, in my opinion the manuscript has to be improved.
Here are my suggestions:
Fig. 1 is not clear (at least in the PDF), letters (A, C, D and E) are not on the photos.
In paragraph starting on line 129 it is not clear what is consider control. I suppose that control is in optimum metal concentration and that is compared with high metal content of a specific trial. However, in comparing A. arenosa with A. halleri this does not seem the case.
Lines 139-140 what does it mean “average zinc content compared to plants growing under normal conditions”? What is the difference between normal conditions (how much Zn?) and average Zn content (how much Zn?)
Line 156 what does it mean Ta-NM and Te-NM?
Not clear sentence. Lines 162.163 Studies conducted on N. caerulescens have shown that plants maintain in mesophyll cells non-toxic for PSII concentrations of Zn and Cd.
Lines 156-173 there is not a link with what observed in A. arenosa
Lines 183-185 This sentence has to be rewritten
Lines 215-216 This sentence need to be cited. There are many ZIP transporters involved in uptake, and for many of them a precise role has not yet been identified.
Lines 204-256 This long paragraph about metal transporters is off-topic. If there are no data for A. arenosa the paragraph must be deleted.
Table 1 reports a list of transporters that have been studied in A.halleri, N. caerulescens but no in A. arenosa. It has nothing to do with the review about A. arenosa.
Author Response
Comments and Suggestions for Authors
The review “The Different Faces of Arabidopsis arenosa – a Plant Species for a Special Purpose” by Gieroń et al., reports exhaustive studies on A. arenosa , a metal hyperaccumulator species that up to now received little interest if compared with A. halleri and N. caerulescens. The work is interesting and add knowledge to the metal hyperaccumulator species. However, in my opinion the manuscript has to be improved.
Here are my suggestions:
Comment 1: Fig. 1 is not clear (at least in the PDF), letters (A, C, D and E) are not on the photos.
Response: Figure 1 and it’s legend was corrected according to Reviewer comment (see lines: 49 in improved manuscript (MS)).
Comment 2: In paragraph starting on line 129 it is not clear what is consider control. I suppose that control is in optimum metal concentration and that is compared with high metal content of a specific trial. However, in comparing A. arenosa with A. halleri this does not seem the case.
Response: The sentence was improved according to Reviewer comment (see lines 130-148 in improved MS).
Comment 3: Lines 139-140 what does it mean “average zinc content compared to plants growing under normal conditions”? What is the difference between normal conditions (how much Zn?) and average Zn content (how much Zn?)
Response: The sentence was improved according to Reviewer comment (see lines 140-142 in improved MS). The missing information was added.
Comment 4: Line 156 what does it mean Ta-NM and Te-NM?
Response: The names of populations mentioned by Reviewer are misleading. We improved the sentence (see lines 150-156 in improved MS).
Comment 5: Not clear sentence. Lines 162.163 Studies conducted on N. caerulescens have shown that plants maintain in mesophyll cells non-toxic for PSII concentrations of Zn and Cd.
Response: We agree with Reviewer, the sentence was not clear so we improve it (see lines 161-163 in improved MS).
Comment 6: Lines 156-173 there is not a link with what observed in A. arenosa
Response: The part of text pointed by Reviewer was improved (see lines 154-177 in improved MS).
Comment 7: Lines 183-185 This sentence has to be rewritten
Response: We agree with Reviewer, the sentence was improved (see lines 180-183 in improved MS).
Comment 8: Lines 215-216 This sentence need to be cited. There are many ZIP transporters involved in uptake, and for many of them a precise role has not yet been identified.
Response: The citation was added according to Reviewer’s comment (see lines 211 in improved MS).
Comment 9: Lines 204-256 This long paragraph about metal transporters is off-topic. If there are no data for A. arenosa the paragraph must be deleted.
Response: We disagree with the Reviewer. In our opinion, uptake and transport of HMs in well-known plant hyperaccumulators as A. halleri and N. caerulescence could be interesting for a typical reader of Plants, who may not have this knowledge yet. Especially when we compare this knowledge with a lack of it for A. arenosa. We think that this paragraph is important and shows the difference between scientific knowledge on A. halleri and A. arenosa.
Comment 10: Table 1 reports a list of transporters that have been studied in A.halleri, N. caerulescens but no in A. arenosa. It has nothing to do with the review about A. arenosa.
Response: We try to show the potential for future studies, which is also the role of the review article. For the time being, there is no data on transporters in A. arenosa, but Preite et al. (2019) mentioned the group of candidate-transporters, which may be involved in Cd and Zn transport in A. arenosa. We believe that this is essential knowledge and point the direction in research in future.
Reviewer 2 Report
The review summarizes the literature in relation to Arabidopsis arenosa cytogenetics, accumulation of heavy metal or impact on photosynthesis. Authors have detailed the current level of understanding in relation of A. arenosa and HM, especially Cd and Zn.
However, authors should clearly identified major research questions, or future research needs, which are key aspects. Explain what this review is bringing new. Emphasize on what we learnt from the studies the authors listed and clearly identify what are the gaps in these understanding. Clearly explain what should be done to fill the gaps and how this knowledge would add to understanding the A. arenosa adaptation to HM stress. Thus, provide opinions and perspectives, otherwise the paper remains simply a literature inventory.
*Conclusions: please conclude with more focus on the major outcomes of the knowledge on A. arenosa and how exactly the understanding of this species adaptation would be informative and useful for science in general and further applications. Avoid arguments such as the knowledge would be an ”interesting issue”.
Metal tolerance and interaction with soil microorganisms should be developed and give more details on the conditions tested and results and nevertheless opinions.
Please shorten the phrases and revised English throughout the entire ms.
Other minor aspects that should be revised:
Lines 87-88
Use of thesis does not seem right.
“Results of a recent study support the thesis that it is possible to overcome the adaptation obstacles caused by reduced selection efficiency, due to increased mutational input [32,33].”
Suggestion “Results of a recent study support the possibility that increased mutational input can overcome the adaptation obstacles caused by reduced selection efficiency.”
Re-phrase sentences in lines 125-127, not clear what study 3 did. Re-phrase 183-186, 191-193, 240-242,
Shorter phrases/sentences should be used, not only in lines 125-127, but also throughout the paper.
Define Ta-NM and Te-NM in line 156.
Line 166 – there is probably extra an space between mechanism and [55].
Line 169 – Cd or any other metal cannot accumulate in the vein. The vasculature is not a storage tissue for any mineral, therefore authors should rephrase. Please explain why is rice relevant for A. arenosa?
Line 211 - Replace “constituents”
Line 215 – specify plant species in which ZIP 19 and ZIP23 play important role in uptake. There is abrupt jump from non hyperacumulator species to hyperacumulators (lines 215-218). Clearly state to what ”genes encoding membrane transporters” authors refer in line 218 – ref. 63 and specify to what plant species they refer to.
Rephrase in line 216 – ”In plants that are not hyperaccumulators” – change with ”In non hyperaccumulator species …” However authors should explain if they refer to metal sensitive species or metal tolerant species but which are non hyperaccumulator.
Clearly specify from the very beginning the plant species authors refer when they use the transporters name (Arabidopsis thaliana?) e.g. ZIP19, ZIP23, IRT1 and so on.
Line 288 – define PCR-DGGE and clearly specify what are those “interesting results”.
Figures
Labeling A-F of fig 1 and A-B in Fig 2 is not visible. Arrows are moved in fig 2 therefore not in the right position.
Fig. 3 Please use anthocyanins not “Anth” in the fig not only in the fig legend. “Anth” in the fig can easily misread as anther.
I suggest adapting fig.3 so that it corresponds to the text.
E.g. lines 129-132 – nor the 2.5 times lower length of the leaves or the size of the roots differs in plants shown in Fig. 3 that the authors are indicating.
Consistency with spelling some words:
e.g. pseudo-metallophytes (line 10) vs pseudometallophyte (line 99) use pseudo-metallophytes
Bibliography
Please revisit and make sure the spelling and format is right. E.g. Arabiopsis halleri is with capital letter Arapbidopsis Halleri (ref. 54), same with Noccaea Caerulescens ref. 50, 52, 55 and so on, Arabidopsis arenosa in ref. 4 and so on.
Author Response
The review summarizes the literature in relation to Arabidopsis arenosa cytogenetics, accumulation of heavy metal or impact on photosynthesis. Authors have detailed the current level of understanding in relation of A. arenosa and HM, especially Cd and Zn.
However, authors should clearly identified major research questions, or future research needs, which are key aspects. Explain what this review is bringing new. Emphasize on what we learnt from the studies the authors listed and clearly identify what are the gaps in these understanding. Clearly explain what should be done to fill the gaps and how this knowledge would add to understanding the A. arenosa adaptation to HM stress. Thus, provide opinions and perspectives, otherwise the paper remains simply a literature inventory.
*Conclusions: please conclude with more focus on the major outcomes of the knowledge on A. arenosa and how exactly the understanding of this species adaptation would be informative and useful for science in general and further applications. Avoid arguments such as the knowledge would be an ”interesting issue”.
Response: The conclusions are rewritten according to the Reviewer’s suggestion.
Metal tolerance and interaction with soil microorganisms should be developed and give more details on the conditions tested and results and nevertheless opinions.
Response: The section was improved according to the Reviewer’s suggestion (see lines 267-280 in improved MS).
Please shorten the phrases and revised English throughout the entire ms.
Response: The MS was checked and phrases were shortened.
Other minor aspects that should be revised:
Lines 87-88
Use of thesis does not seem right.
“Results of a recent study support the thesis that it is possible to overcome the adaptation obstacles caused by reduced selection efficiency, due to increased mutational input [32,33].”
Suggestion “Results of a recent study support the possibility that increased mutational input can overcome the adaptation obstacles caused by reduced selection efficiency.”
Response: The sentence was improved according to the Reviewer’s comment (see line 91 in improved MS).
Re-phrase sentences in lines 125-127, not clear what study 3 did. Re-phrase 183-186, 191-193, 240-242. Shorter phrases/sentences should be used, not only in lines 125-127), but also throughout the paper.
Response: All mentioned sentences were re-phrased.
Define Ta-NM and Te-NM in line 156.
Response: The sentence was changed and abbreviations were removed (See lines: 151-158 in improved MS).
Line 166 – there is probably extra an space between mechanism and [55].
Response: The extra space was removed.
Line 169 – Cd or any other metal cannot accumulate in the vein. The vasculature is not a storage tissue for any mineral, therefore authors should rephrase. Please explain why is rice relevant for A. arenosa?
Response: We cannot agree with the Reviewer. We refer to the paper by Morina and Küpper (2020), in which authors clearly stated as follow:
“On the contrary, in the Cd treatments, especially in Cd75, the trichomes were not visible on the leaves. In the Cd15 treatment, Zn accumulation increased dramatically, in particular reaching a new accumulation maximum in the bundle sheath cells (Fig. 8). In the Cd75 treatment, Zn accumulation in the bundle sheath decreased again but accumulation started to appear inside the veins. Cadmium, on the other hand, was rather evenly distributed within the leaf in the Cd15 treatment, except for an enhanced accumulation in the main vein, which directly correlated with a severely decreased ΦEt2o in mesophyll cells on top of the main vein (Fig. 8). In the Cd75 treatment, Cd started to accumulate in the smaller veins, exceeding the accumulation in the mesophyll cells.”
We agree with the Reviewer’s comment on rice, the sentence was removed.
Line 211 - Replace “constituents”
Response: The sentence was changed (See line 205 in improved MS).
Line 215 – specify plant species in which ZIP 19 and ZIP23 play important role in uptake. There is abrupt jump from non hyperacumulator species to hyperacumulators (lines 215-218). Clearly state to what ”genes encoding membrane transporters” authors refer in line 218 – ref. 63 and specify to what plant species they refer to.
Response: All the discussed genes were tested for N. caerulescence and A. halleri, as shown in Table 1.
Rephrase in line 216 – ”In plants that are not hyperaccumulators” – change with ”In non hyperaccumulator species …” However authors should explain if they refer to metal sensitive species or metal tolerant species but which are non hyperaccumulator.
Response: The section was improved (See lines 209-213 in the improved MS).
Clearly specify from the very beginning the plant species authors refer when they use the transporters name (Arabidopsis thaliana?) e.g. ZIP19, ZIP23, IRT1 and so on.
Response: The section was improved (See lines in improved MS). Most of the discussed genes concerned A. halleri and N. caerulescence, if it was a different plant species, we immediately entered the information in the text.
Line 288 – define PCR-DGGE and clearly specify what are those “interesting results”.
Response: The sentence was improved according to the Reviewer comment (See lines 278-280 in improved MS).
Figures
Labeling A-F of fig 1 and A-B in Fig 2 is not visible. Arrows are moved in fig 2 therefore not in the right position.
Fig. 3 Please use anthocyanins not “Anth” in the fig not only in the fig legend. “Anth” in the fig can easily misread as anther.
I suggest adapting fig.3 so that it corresponds to the text.
E.g. lines 129-132 – nor the 2.5 times lower length of the leaves or the size of the roots differs in plants shown in Fig. 3 that the authors are indicating.
Response: All figures are improved and corrected.
Consistency with spelling some words:
e.g. pseudo-metallophytes (line 10) vs pseudometallophyte (line 99) use pseudo-metallophytes
Response: The MS was corrected according to the Reviewer’s suggestion.
Bibliography
Please revisit and make sure the spelling and format is right. E.g. Arabiopsis halleri is with capital letter Arapbidopsis Halleri (ref. 54), same with Noccaea Caerulescens ref. 50, 52, 55 and so on, Arabidopsis arenosa in ref. 4 and so on.
Response: Bibliography was corrcted.
Reviewer 3 Report
From what I've read this manuscript seems to be very good,
Title of manuscript to accept
Abstract
This species is common in Poland
Part 1
Is this species an indicator plant ?
Part 2
Correct
Part 3
- 141 - non contamination ... ?
L 149 - You can see paper on resistance of photosynthetic apparatus ....
For instance::
Bączek-Kwinta R., Juzoń K., Borek M., Antonkiewicz J. 2019. Photosynthetic response of cabbage in cadmium-spiked soil. Photosynthetica, 57, 3, 731-739. DOI: 10.32615/ps.2019.070
Conclusion
Please add information where I can use this plant to remediation ? Mabe: mine dump and other places ....

Author Response
Comments and Suggestions for Authors
Comment 1: From what I've read this manuscript seems to be very good,
Lines 2-3 Title of manuscript to accept
Response: We thank the Reviewer for the opinion.
Comment 2: Abstract
Line 8 This species is common in Poland
Response: We thank the Reviewer for the comment. The sentence was improved (see line 9 in improved manuscript (MS)).
Comment 3: Part 1
Line 40 Is this species an indicator plant ?
Response: A. arenosa is not an indicator plant.
Comment 4: Part 2
Line 63 Correct
Response: The sentence was corrected according to Reviewer comment (see lines 68-71 in improved MS).
Comment 5: Part 3
Line 141 - non contamination ... ?
Response: The sentence was improved (see lines 140-143 in improved MS).
Comment 6: Lines 149-151 - You can see paper on resistance of photosynthetic apparatus ....
For instance::
Bączek-Kwinta R., Juzoń K., Borek M., Antonkiewicz J. 2019. Photosynthetic response of cabbage in cadmium-spiked soil. Photosynthetica, 57, 3, 731-739. DOI: 10.32615/ps.2019.070
Response: Unfortunately, the sentence is about photosynthesis in hyperaccumulators of Cd and Zn, however, cabbage is not a hyperaccumulator. Nevertheless, we have included mentioned paper to references (see line 151 in improved MS).
Comment 7: Conclusion
Please add information where I can use this plant to remediation? Mabe: mine dump and other places ....
Response: In comparison with A. halleri, biomass of A. arenosa on metalliferous soils is lower, so the potential of phytoremediation is also lower. Moreover, there is a lack of data which may characterize the possibility of phytoremediation using A. arenosa.
Reviewer 4 Report
Dear authors,
A very interesting and thorough review article - I very much enjoyed reading its contents.
My two main concerns are;
(1) numerous English language issues throughout the manuscript (please see the annotated attachment where I have identified each concern) which must be corrected.
(2) The lack of annotation of Figures 1 and 2. Both of these two Figures need to be improved (Namely the addition of panel labelling).
These are my only concerns so please address these issues in order to improve the impact and quality of your submitted review article.
Regards,
Andrew.

Author Response
Comments and Suggestions for Authors
Comment 1: A very interesting and thorough review article - I very much enjoyed reading its contents. My two main concerns are;
(1) numerous English language issues throughout the manuscript (please see the annotated attachment where I have identified each concern) which must be corrected.
Response: We thank very much the Reviewer for suggested improvements.
Comment 2: The lack of annotation of Figures 1 and 2. Both of these two Figures need to be improved (Namely the addition of panel labelling).These are my only concerns so please address these issues in order to improve the impact and quality of your submitted review article.
Response: Both figures were improved according to the Reviewer comment (see lines 49 and 63 in the improved manuscript (MS)).
Comment 3: Line 10 of this Arabidopsis species
Response: The sentence was improved (see line 12 in the improved MS).
Comment 4: Lines 11-12 sentence wording does not make sense - needs to be rewritten
Response: The sentence was improved (see line 12-15 in the improved MS).
Comment 5: Line 15 these two
Response: The sentence was improved (see line 17 in the improved MS).
Comment 6: Line 24 is this accepted terminology?
Response: Yes, the terminology is correct.
Comment 7: Line 25 are the result of
Response: The sentence was improved (see line 29 in the improved MS).
Comment 8: Line 26 via
Response: The sentence was improved (see line 30 in the improved MS).
Comment 9: Line 26 more
Response: The sentence was improved (see line 30 in the improved MS).
Comment 10: Line 32 subfertile?
Response: The sentence was improved (see line 37 in the improved MS).
Comment 11: Line 32 has been
Response: The sentence was improved (see line 37 in the improved MS).
Comment 12: Line 40 is a
Response: The sentence was improved (see line 44-46 in the improved MS).
Comment 13: Line 40 delete the 's'
Response: The sentence was improved (see line 44 in the improved MS).
Comment 14: Line 41 species
Response: The sentence was improved (see line 44-46 in the improved MS).
Comment 15: Line 41 environments which are
Response: The sentence was improved (see line 44-46 in the improved MS).
Comment 16: Line 48 The five panel which make up Figure 1 need to be clearly labeled A, B, C, D and E so that the reader can readily determine which Figure panel contains which species.
Response: The figure was improved according to the Reviewer suggestion (see line 49 in the improved MS).
Comment 17: Line 52 the ground level, and no runners form from the rosette.
Response: The sentence was improved (see line 54-55 in the improved MS).
Comment 18: Line 57 this needs rewording for clarity
Response: The sentence was improved (see lines 59-62 in the improved MS).
Comment 19: Line 60 Each Figure 2 panel needs to be labeled (A and B), and if the yellow coloured arrows are to remain then they must be clearly explained in the Figure 2 legend.
Response: The Figure 2 was corrected and improved by adding two new pictures (see line 63 in the improved MS).
Comment 20: Line 65 , most importantly, avoid
Response: The sentence was improved (see lines 70 in the improved MS).
Comment 21: Line 67 pairings
Response: The sentence was improved (see lines 72 in the improved MS).
Comment 22: Line 70 word choice?
Response: The sentence was improved (see lines 75 in the improved MS).
Comment 23: Lines 72-73 are also involved in other functions
Response: The sentence was improved (see lines 77 in the improved MS).
Comment 24: Line 74 do not use 'etc'. You need to be specific in scientific writing
Response: The sentence was improved (see lines 78 in the improved MS).
Comment 25: Line 75 which
Response: The sentence was improved (see lines 79 in the improved MS).
Comment 26: Line 82 , that is;
Response: The sentence was improved (see lines 85 in the improved MS).
Comment 27: Line 84 of the
Response: The sentence was improved (see lines 87 in the improved MS).
Comment 28: Line 87 , and thus autotetraploidyzation,
Response: The sentence was improved (see lines 90 in the improved MS).
Comment 29: Line 92 the
Response: The sentence was improved (see lines 95 in the improved MS).
Comment 30: Line 99 , that is,
Response: The sentence was improved (see lines 102 in the improved MS).
Comment 31: Line 100 inhabits both
Response: The sentence was improved (see lines 102 in the improved MS).
Comment 32: Line 102 zinc (Zn) and low cadmium (Cd)
Response: The sentence was improved (see lines 105 in the improved MS).
Comment 33: Line 104 hyperaccumulate Zn
Response: The sentence was improved (see lines 107 in the improved MS).
Comment 34: Line 105 Van der Ent and colleagues (2013)
Response: The sentence was improved (see lines 107-108 in the improved MS).
Comment 35: Line 109 for their accumulation
Response: The sentence was improved (see lines 110 in the improved MS).
Comment 36: Line 109 observable
Response: The sentence was improved (see lines 111 in the improved MS).
Comment 37: Line 109 being displayed
Response: The sentence was improved (see lines 111 in the improved MS).
Comment 38: Line 110 Zn
Response: The sentence was improved (see lines 112 in the improved MS).
Comment 39: Lines 111-113 sentence wording requires attention - needs to be rewritten
Response: The sentence was improved (see lines 113-115 in the improved MS).
Comment 40: Line 115 the Brassicaceae family
Response: The sentence was improved (see lines 116 in the improved MS).
Comment 41: Line 117 the
Response: The sentence was improved (see lines 118 in the improved MS).
Comment 42: Line 120 you need to explain this more clearly to the reader
Response: The sentence was improved (see lines 120-121 in the improved MS).
Comment 43: Line 122 hyperaccumulate
Response: The sentence was improved (see lines 123 in the improved MS).
Comment 44: Line 122 delete
Response: The sentence was improved (see lines 123 in the improved MS).
Comment 45: Lines 122-125 wording requires attention. I also suggest that this is broken into two sentences for reader clarity.
Response: The sentence was improved (see lines 123-128 in the improved MS).
Comment 46: Line 128 the waste of a proximal lead (Pb) and Zn smelter
Response: The sentence was improved (see lines 129 in the improved MS).
Comment 47: Line 129 habitats
Response: The sentence was improved (see lines 130 in the improved MS).
Comment 48: Line 130 (Cu)
Response: The sentence was improved (see lines 131 in the improved MS).
Comment 49: Line 134 Zn/Pb
Response: The sentence was improved (see lines 134 in the improved MS).
Comment 50: Line 141 Zn
Response: The sentence was improved (see lines 141 in the improved MS).
Comment 51: Line 146 delete
Response: The sentence was improved (see lines 148 in the improved MS).
Comment 52: Line 149 general,
Response: The sentence was improved (see lines 149 in the improved MS).
Comment 53: Line 149 delete the 's'
Response: The sentence was improved (see lines 149 in the improved MS).
Comment 54: Line 156 the
Response: The sentence was improved (see lines 154 in the improved MS).
Comment 55: Line 163 wording does not make sense - this section needs to be rewritten
Response: The section was improved (see lines 160-166 in the improved MS).
Comment 56: Line 164 full name on first mention
Response: The sentence was improved (see line 163 in the improved MS).
Comment 58: Line 166 delete
Response: The sentence was removed.
Comment 59: Line 166 (rice)
Response: The sentence was removed.
Comment 60: Line 176 population: the HM hyperaccumulator.
Response: The sentence was improved (see line 173 in the improved MS).
Comment 61: Line 176 total chlorophyll? As in the above sentence you state Chl a is the same in arenosa and halleri
Response: The sentence was improved (see line 175 in the improved MS).
Comment 62: Line 183 has been demonstrated in A. thaliana that
Response: The sentence was improved (see line 180 in the improved MS).
Comment 63: Lines 184-186 some words seem to be missing here linking these two parts of the sentence together?
Response: The sentence was improved (see lines 180-181 in the improved MS).
Comment 64: Line 190 sites
Response: The sentence was improved (see line 186 in the improved MS).
Comment 65: Line 201 are characterized
Response: The sentence was improved (see line 196 in the improved MS).
Comment 66: Line 203 heavy metals
Response: The sentence was improved (see line 198 in the improved MS).
Comment 67: Line 205 HM
Response: The sentence was improved (see line 204 in the improved MS).
Comment 68: Line 207 delete
Response: The sentence was improved (see line 207 in the improved MS).
Comment 70: Line 216 plant species
Response: The sentence was improved (see line 209 in the improved MS).
Comment 71: Line 217 In addition,
Response: The sentence was improved (see line 211 in the improved MS).
Comment 72: Line 219 full name on first mention
Response: The sentence was improved (see line 214 in the improved MS).
Comment 73: Line 220 and/or nickel (Ni)
Response: The sentence was improved (see line 214 in the improved MS).
Comment 74: Line 221 expression
Response: The sentence was improved (see line 215 in the improved MS).
Comment 77: Line 228 is mediated by the HM transporter, the HMA4 protein, a plasma membrane ATPase
Response: The sentence was improved (see line 221 in the improved MS).
Comment 78: Lines 229-230 sentence wording is poor - requires attention
Response: The sentence was improved (see lines 221-224 in the improved MS).
Comment 79: Line 232 why italics when discussing a protein? Proteins should be upright capitals, whereas genes and transcripts in capital italics
Response: The sentence was improved (see line 224 in the improved MS).
Comment 80: Line 235 , including
Response: The sentence was improved (see line 227 in the improved MS).
Comment 81: Line 238 full name
Response: The sentence was improved (see line 229 in the improved MS).
Comment 82: Line 240 organs of a plant, they enter ...
Response: The sentence was improved (see line 231 in the improved MS).
Comment 83: Lines 241-242 wording?
Response: The sentence was improved (see lines 231-233 in the improved MS).
Comment 84: Line 243 full name
Response: The sentence was improved (see line 234 in the improved MS).
Comment 85: Line 243 the
Response: The sentence was improved (see line 234 in the improved MS).
Comment 86: Line 252 , such as ... and ...
Response: The sentence was improved (see line 242 in the improved MS).
Comment 87: Line 253 the NRAMP gene family
Response: The sentence was improved (see line 243 in the improved MS).
Comment 89: Line 257 putatively assigned functional roles in ....
Response: The sentence was improved (see line 247 in the improved MS).
Comment 90: Line 260 wording?
Response: The sentence was improved (see line 252 in the improved MS).
Comment 91: Line 261 full name and italics
Response: The sentence was improved (see line 253 in the improved MS).
Comment 92: Line 280 were determined to have a
Response: The sentence was improved (see line 271 in the improved MS).
Comment 93: Line 293 while also existing as a
Response: The sentence was improved (see line 285 in the improved MS).
Round 2
Reviewer 2 Report
The manuscript has been revised; however, there are still minor issues that should be addressed.
Line 125 There is a reference left-(Nadgórska-Socha et al., 2015). Only [48] should be left.
Line 130: 133 “However, the data contained in the paper presents only the high percentage of plants that survived in soil highly contaminated by the waste of a proximal lead (Pb) and Zn smelter [3]”. Please explain what “only high percentage of plants that survived in soil highly contaminated” means.
Still the phrases are long and hard to follow. In section 3 phrases are still too complicated and hard to follow. Please avoid using: “it was shown, it was demonstrated, it was observed”. All these are obvious; otherwise, authors would not have cited the work. They only add more words and make the sentences/phrases longer and harder to follow. These “it was shown, it was demonstrated, it was observed”are extremely numerous throughout the entire ms.
One example in section 3.
“Similarly, smaller plant sizes were observed for the A. arenosa population growing on a Zn/Pb waste heap, and the leaves of the plants were thicker, narrower, and had fewer trichomes.”
Proposed
Similarly, A. arenosa population growing on a Zn/Pb waste heap had smaller size, thicker, and narrower leaves, with fewer trichomes.
“Moreover, the performed root test showed a higher level of tolerance to Cd, Zn and Pb for the population from the heap compared to reference population [18]. On the other hand, it was demonstrated that, both the metallicolous (M) and non-metallicolous (NM) tetraploid A. arenosa plants growing in hydroponic solution without HMs were characterised by a displayed/had higher above-ground biomass production compared to the A. halleri plants [22].”
I suggest continuing here with the A. arenosa rather than introducing Nc.
“However, in the case of increasing Cd content in the medium, in Cd containing media, a more significant decrease in biomass was observed in the NM population than in the M population of A. arenosa [22]. …..”
I stop here, but the entire section should be re-written.
The whole paragraph on metal transporters involved in tolerance is on other species than A. arenosa. It is fine as it describes the context. But, the paragraph on A. arenosa (lines 270-274) that would have been the main interest in this paper says very little without deeper explanation. Those 5 candidate genes should be named and discuss in the context to all aspects previously described. And here, there is place for opinions, identifying the gaps, etc. Authors should explain what are the mechanisms of HM resistance in M Aa populations based on what has been fount for this species and other hyperaccumulator species Ah, Nc etc.
Conclusions have been re-written and indeed include now some opinion.
Please remove the last phrases in the conclusions. Conclusions should not include reference to Figs. In the text. Also, please rephrase “The above knowledge will contribute to the possibility of using the A. arenosa in the phytoremediation process”, as it is very unlikely that the hyperaccumulator species such as Ah, Nc, and even Aa will ever be used in phytoremediation due to their low biomass. The knowledge we gather by investigating their tolerance mechanisms can be further exploited in high biomass plant species that could be used for phytoremediation purposes.
Author Response
The manuscript has been revised; however, there are still minor issues that should be addressed.
Comment 1
Line 125 There is a reference left-(Nadgórska-Socha et al., 2015). Only [48] should be left.
Response:
Reference was removed (see line 124 in the improved manuscript).
Comment 2
Line 130: 133 “However, the data contained in the paper presents only the high percentage of plants that survived in soil highly contaminated by the waste of a proximal lead (Pb) and Zn smelter [3]”. Please explain what “only high percentage of plants that survived in soil highly contaminated” means.
Response:
The text was modified (see lines 128-131 in the improved manuscript)
Comment 3
Still the phrases are long and hard to follow. In section 3 phrases are still too complicated and hard to follow. Please avoid using: “it was shown, it was demonstrated, it was observed”. All these are obvious; otherwise, authors would not have cited the work. They only add more words and make the sentences/phrases longer and harder to follow. These “it was shown, it was demonstrated, it was observed” are extremely numerous throughout the entire ms.
Response:
The text in section 3 was modified. The number of phrases : “it was shown, it was demonstrated, it was observed” was reduced. Also the length of phrases was decreased, where it was possible (see section 3 text in brown fonts).
Comment 4
One example in section 3.
“Similarly, smaller plant sizes were observed for the A. arenosa population growing on a Zn/Pb waste heap, and the leaves of the plants were thicker, narrower, and had fewer trichomes.”
Proposed
Similarly, A. arenosa population growing on a Zn/Pb waste heap had smaller size, thicker, and narrower leaves, with fewer trichomes.
Response:
The sentence was modified according to reviewer suggestion (see lines 136-137 in the improved manuscript).
Comment 5
“Moreover, the performed root test showed a higher level of tolerance to Cd, Zn and Pb for the population from the heap compared to reference population [18]. On the other hand, it was demonstrated that, both the metallicolous (M) and non-metallicolous (NM) tetraploid A. arenosa plants growing in hydroponic solution without HMs were characterised by a displayed/had higher above-ground biomass production compared to the A. halleri plants [22].”
Response
The sentence was modified according to reviewer suggestion (see lines 138-142 in the improved manuscript).
Comment 6
I suggest continuing here with the A. arenosa rather than introducing Nc.
Response
The sentence was removed according to reviewer suggestion (see lines 142-143 in the improved manuscript).
Comment 7
“However, in the case of increasing Cd content in the medium, in Cd containing media, a more significant decrease in biomass was observed in the NM population than in the M population of A. arenosa [22]. …..”
Response:
The sentence was modified according to reviewer suggestion (see lines 142-143 in the improved manuscript).
Comment 8
I stop here, but the entire section should be re-written.
Response:
The section 3 was rewritten, see the text in blue and brown fonts.
Comment 9
The whole paragraph on metal transporters involved in tolerance is on other species than A. arenosa. It is fine as it describes the context. But, the paragraph on A. arenosa (lines 270-274) that would have been the main interest in this paper says very little without deeper explanation. Those 5 candidate genes should be named and discuss in the context to all aspects previously described. And here, there is place for opinions, identifying the gaps, etc. Authors should explain what are the mechanisms of HM resistance in M Aa populations based on what has been fount for this species and other hyperaccumulator species Ah, Nc etc.
Response:
The paragraph on A. arenosa was modified and extended (see lines 261-279 in the improved manuscript).
Comment 10
Conclusions have been re-written and indeed include now some opinion.
Response:
Thank you very much.
Comment 11
Please remove the last phrases in the conclusions. Conclusions should not include reference to Figs in the text.
Response:
The last phrases in the conclusions were removed (see Conclusions in the improved manuscript).
Comment 12
Also, please rephrase “The above knowledge will contribute to the possibility of using the A. arenosa in the phytoremediation process”, as it is very unlikely that the hyperaccumulator species such as Ah, Nc, and even Aa will ever be used in phytoremediation due to their low biomass. The knowledge we gather by investigating their tolerance mechanisms can be further exploited in high biomass plant species that could be used for phytoremediation purposes.
Response:
The sentence was modified according to reviewer suggestion (see lines 325-327 in the improved manuscript).